# CROSS-MODAL OBJECT IDENTIFICATION USING VISION AND TACTILE SENSING

## ABSTRACT

Combining information from different sensory modalities to execute goal directed actions is a key aspect of human intelligence. Specifically, human agents are very easily able to translate the task communicated in one sensory domain (say vision) into a representation that enables them to complete this task when they can only sense their environment using a separate sensory modality (say touch). In order to build agents with similar capabilities, in this work we consider the problem of a retrieving a target object from a drawer. The agent is provided with an image of a previously unseen object and it explores objects in the drawer using only tactile sensing to retrieve the object that was shown in the image without receiving any visual feedback. Success at this task requires close integration of visual and tactile sensing. We present a method for performing this task in a simulated environment using an anthropomorphic hand. We hope that future research in the direction of combining sensory signals for acting will find the object retrieval from a drawer to be a useful benchmark problem.

## 1 INTRODUCTION

A core aspect of human intelligence is the ability to integrate and translate information between multiple sensory modalities to achieve an end goal. For example, we have no trouble discriminating between a set of keys, a wallet or a watch kept in our pocket by simply feeling them with our hands. Similarly, we can easily retrieve a desired object present inside a dark drawer even if we can't see the objects using touch sensation from our hands. Not only can we retrieve a previously known object, but if we are shown an image of a previously unseen object, we would still have no trouble retrieving this object using only tactile exploration inside the drawer even in absence of any visual feedback. Such translation of information between sensory modalities is not specific to tactile and vision, but is noticed between other modalities as well. For instance, it is easy to imagine someone walking down the stairs and opening the door by simply hearing the sound that was generated. These examples demonstrate how easily humans can translate information between sensory modalities.

Different sensory modalities provide a different view of the same underlying reality. The ability to transform between sensory modalities therefore provides an interesting way to learn useful representations of sensory inputs. Recent work in self-supervised learning has made extensive use of this observation and shown that useful visual features can be learned by predicting, from images, corresponding sounds (Owens et al., 2016), ego-motion (Agrawal et al., 2015; Jayaraman & Grauman, 2015), depth or even predicting color values from grayscale images (Zhang et al., 2016).

In addition to learning feature representations, another and possibly more critical use of sensing from multiple modalities is performing goal directed actions in partially observable settings. In the running example of retrieving objects from a drawer, the agent receives only the image of the object as input and in absence of any light source in the drawer, the agent solely relies on its tactile sensing to find the object. Other examples are a pedestrian getting alerted when she hears the sound of a car coming from the back or animals in the jungle being alerted of a tiger behind the bushes by the sound of the movement. Yet another example showing close integration of two modalities (vision and touch) is a study that found it became almost impossible for human participants to perform the seemingly trivial task of picking up a matchstick and lighting it when their hands were anesthetized (Johansson & Flanagan, 2009).

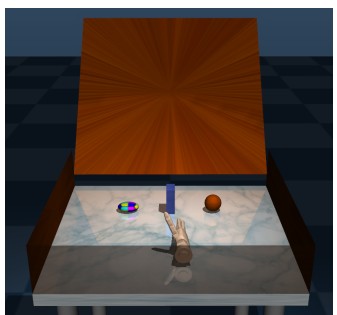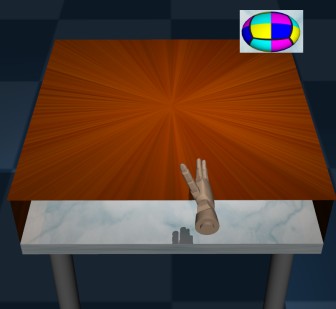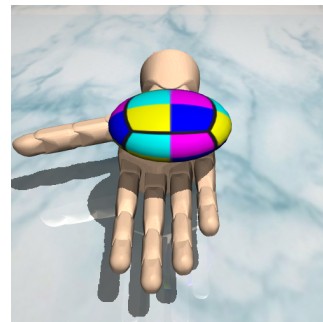

Figure 1: (*Left*) Shows our experimental setup. $p$ objects are in a drawer and a dexterous hand equipped with tactile sensing can explore novel objects using deterministic routines. In this case, $p = 3$ but we compared performance by varying the number of objects (*Middle*) We are presented with a query image as seen by the inset in the top right of the image. We explore the objects in the drawer using tactile sensing only to identify the object (*Right*) We then retrieve the object by applying a grasping routine

In this work we use the task of retrieving objects from a drawer as an experimental setup to investigate joint learning from two sensory modalities of vision and touch. Because the agent is provided only with a visual image of the object to be retrieved, it must translate into the representation space of tactile sensing to retrieve objects only by touching them. In the general case of retrieving the object, the agent must first explore spatially to locate where the objects are. Once it finds the object, it must move its fingers in an exploratory manner to collect information required to determine if the object is the one that needs to be retrieved. Solving this problem in its full generality requires not only good goal directed exploration strategies and also a method for translating between different sensory signals. We therefore think that object retrieval from a drawer is a good challenge problem for investigating different models that combine visual and tactile information for a target end task.

In our setup the agent learns a mapping from visual to tactile signals using unsupervised exploration. This mapping enables the agent to determine the representation of the image in the representation space of tactile sensing (i.e. expected tactile response). The agent explores each object present in the drawer by touching it and compares the result of its exploration with the expected tactile response. Performing this comparisons requires a good representation of raw tactile signals. For learning such a representation, we leverage the results in image classification, where it was found that a network pre-trained to classify images from the Imagenet dataset into one thousand image categories learns features useful for many other visual tasks. Similar to image classification, we pose the task of classifying objects from tactile signals collected by touching eleven objects. We show that tactile representation learned by performing the task of classification, generalize and can be used to retrieve novel objects. We present results in a simulated environment and the agent explores the objects using an anthropomorphic hand.

## 2 RELATED WORK

One of the earliest works presenting haptics as a sensory modality to explore the world was by Gibson (Gibson (1962)).Gibson showed that object recognition dramatically decreased when one could not actively interact with an object. Lederman and colleagues [Lederman & Klatzky (1993)]. They describe the various exploratory prcoedures (EP) that humans can perform to understand various object properties such as volume, temperature, friction, etc. Multi-modal learning is a key component for how biological agents learn and build models of objects. It can be argued by looking at failure modes to modern day robotics (rob) that it is exactly this lack in multi-modal learning that requires further study.

Earlier work in haptic exploration includes (Caselli et al. (1994), Goldberg & Bajcsy (1984)) who employed various hand engineered features to recognized objects using haptics. The challenges faced were largely due to robust sensors and the ability to control these sensors to explore objects effectively.

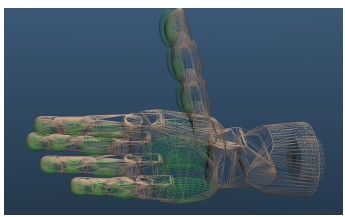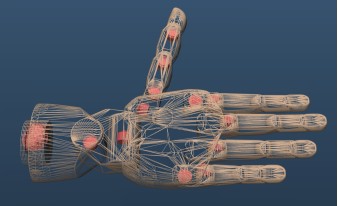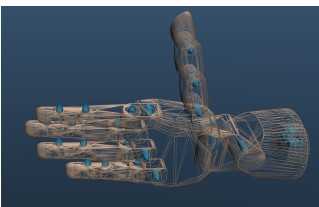

Figure 2: For fine manipulation humans rely mostly on touch, dexterous hands that are equipped with touch sensors could help mimic complex movements(*Left*) Shows the MPL hand with 19 touch sensors depicted in green. (*Middle*) The actuators can be seen in red. (*Right*) Shows the joints that are available in this hand. The hand is under-actuated so the number of joints are greater than the number of actuators

More recently, Chu et al. (Chu et al. (2013)) measure various physical properties of objects using the bio-tac sensor using five different exploration procedures (EP). In addition, they also collect adjectives for each object and the corresponding They then compute precision, recall scores using a static hand-engineered feature and dynamic feature model employing Hidden Markov Models and compute precision, recall scores on a held out dataset. Similarly, Schneider et al. (2009) et al. also classify objects using a bag-of-words appraoch.

Romano et al. (Romano et al. (2011)) mimic human tactile sensing for grasping by hand engineering features that can appropriately measure slippage. They then design a control strategy that can grasp and place that employs the tactile responses. They show that in cases where objects are crushable, a naive controller crushes 100% of the time as compared to a controller that effectivel leverages tactile sensing.

Others, such as Sinapov et al. (Sinapov et al. (2011)) have considered building object representations using vibrotactile sensation. They show that they can classify surface properties using data collected from five different EPs. Similarly, Fishel & Loeb (2012) classify textures using the bio-tac sensor using a Bayesian exploration strategy. While, Gorges et al. (2010) employ a *palpatation sequence* that is not learnt to effectively explore objects in a multi-fingered robot.

Our work relates to work by Gao et al. (2016) who show that combining visual and haptic information can lead to better classification of haptic properties. More recently, Calandra et al. (Calandra et al. (2017)) show that employing tactile inputs into a learnt model can help improve predictions of graspability. (OpenAI: Marcin Andrychowicz (2018)) have shown that tactile features may not be required for certain constrained in-hand manipulation tasks. While this may seem contrary, this in fact is not a representative task. Further, the setup employed by the authors substitutes tactile sensing with a very rich 3D data along with a powerful learning method thus navigating around tactile sensing requirements.

## 3 MODEL

**Task Setup** :
Figure 1 presents our task setup. A subset of objects from Figure 3 are placed in a drawer. An image of the object from a fixed pose is presented to the agent. The agent explores each object using a set of pre-determined routines combining palpation and grasping-like movements. The agent then identifies the object it needs to grasp and executes a grasping routine. In our setup the movement between the objects and grasping, is done using a pre-determined routine.

The object is held translationally fixed in space but can rotate about its axes. The hand is initialized close to the object. The hand is translationally fixed, in that it cannot slide but it can rotate around the wrist joint. The fingers can explore their movements with only the restrictions imposed by the joints themselves. That is the fingers , say, cannot bend backwards towards the wrist.

For each episode of 500 time steps long, the haptic forces $\mathcal{H}_t$ and the corresponding Images $\mathcal{I}_t$ are collected. Each object is presented in multiple random poses. The dataset consists of 500 samples per object, each sample is 500 timesteps long and has 19 dimensions.

**Data Setup:**

We use a simulated model of the anthropomorphic hand used as part of SouthHampton Hand Assesment Procedure (SHAP) test suite built by Light et al. (2002) (see Figure 3). The SHAP procedure was established for evaluating prosthetic hands and arms. With this idea in mind, prior work (**?**) built a prosthetic arm which could theoretically perform all useful human hand movements. Based on this hand and the DARPA Haptix challenge, a simulated model of a similar hand (but with fewer sensors) (Kumar (2016)) was built using the *Mujoco* physics engine (Todorov et al. (2012)). This model was made publicly available and we use this for all our experiments.

The hand has five fingers and 22 joints out of which many are tendon coupled. For example, curling the tip of a finger automatically actuates the other two joints on the finger so that the finger moves towards the palm. Because of these couplings, the resultant dynamics can be quite complex and articulated. Out of the 22 joints, thirteen are actuated. Out of these thirteen, ten joints control the motion of fingers and the other three control the rotation of the hand. Additionally, there are three degrees of motion along the (x, y, z) axis and therefore overall 16 degrees of actuation. The hand is controlled by setting the position of these 16 actuators. In addition, the hand is equipped with 19 contact sensors (as seen in 3) that measure normal contact forces that are computed by *Mujoco*. These sensors form the basis of our tactile sensing.

In our setup, we have two sets of networks. Network $f_1$ accepts as inputs, images at time t , defined by $\mathcal{I}_t$. It then learns to predict the haptic responses for the object being explored defined by $\mathcal{H}_t$. This network is optimized by minimizing the following objective function

$$\min_{f_{1_\theta}} \| f_1(\mathcal{I}_t) - \mathcal{H}_t \|^2 \tag{1}$$

Given, tactile responses can we discriminate a set of objects effectively? To do this, we train a separate network $f_2$. This network accepts, as inputs, $\mathcal{H}_t$ and learns to predict object identities $\mathcal{Y}_t$. We then minimze the cross entropy loss as in 2.

$$\min_{f_{2_\theta}} \sum_i^K \mathcal{Y}_t log(f_2(\mathcal{H}_t)) \tag{2}$$

To simulate how an agent would be able to identify an object during test time we present an image $\mathcal{I}$ to the model. Network $f_1$ predicts the haptic responses to this object - $\hat{\mathcal{H}}$. The predicted haptic responses, $\hat{\mathcal{H}}$ are then used to compute the predicted object category $\hat{\mathcal{Y}}$. We can then apply a learnt grasping routine to grasp the object and retrieve it.

To train the haptics predictor network, $\mathcal{F}_1$ the inputs were gray scaled, 64x64 images $\mathcal{I}_t$ with the focus object centered in the image as seen in Figure 3. The network consisted of three convolutional layers of filters 32, 32, 32. The kernel size was 5,5 for each layer. The output of the convolutional layer was then fed into three sequential fully connected layers of sizes [..., 1024], [1024, 512], and [512, 19] to predict the 19 dimensional haptic forces. The groundtruth predictions were per-channel averaged haptic forces across an entire episode length of time T. We then trained the network using ADAM (Kingma & Ba (2014)) with an initial learning rate set to 1e-4.

To train the object discriminator network, $\mathcal{F}_2$ the inputs were average haptic forces over an entire episode which have 19 dimensions. These inputs were then passed into a network of fully connected layers of sizes [19, 250], [250, 250], and [250, K]. We then minimize the cross entropy loss between the ground truth and predicted object categories using the ADAM with an initial learning rate set to 1e-4.

In both cases performing normalization of the input images and haptic forces was critical to training the network. For the images, a simple mean subtraction and division by the standard deviation sufficed. For the haptic cases since the forces were different across different dimensions and doing a simple normalization that resulted in small values that were outside the range of $tanh$ function resulted in difficulties in good predictions. We introduced a scale term to the normalized output so that distribution of the target data was inline with the output range of the network.

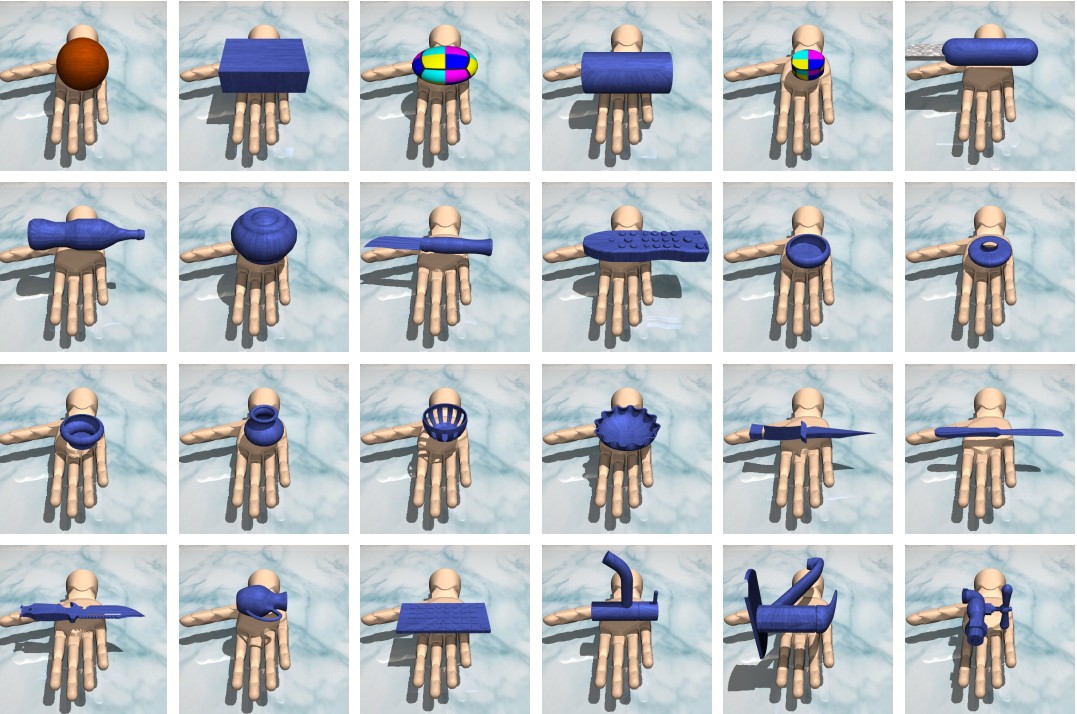

Figure 3: Displays the objects used in our experiments. We used a set of 25 objects. These were imported from the ShapeNet dataset (Chang et al. (2015)). Each object was presented in various different poses. The hand was initialized at the same location for each sample while the object was randomized in each trial.

## 4  RESULTS

We present three sets of experiments in this section. First, we study how hard it is to identify an object using tactile sensing. We do this on novel poses that the model has not seen during training. Next, we explore the question of effective exploration length for these experiments. Finally, we study the problem of identifying novel objects in the dark.

### 4.1  OBJECT IDENTIFICATION WITH NOVEL OBJECT POSES DURING TEST

Before identifying novel objects in the dark, we wished to understand how challenging the problem of identifying an object through tactile sensing was. The inputs in this case were average haptic forces over the entire sampling routine. The training consisted of 400 training samples per object category. Each sample presents the object a random rotation about the z-axis. In total, 4400 were used in training. We used 50 samples per object to evaluate the model. During test time, another 50 samples from each object class but unseen random poses were provided. The model was asked to correctly identify the objects, this classification accuracy is reported in the table 4.1.

For the object identification problem, we compare the classification accuracy of two networks. First, the pretrained $f_2$ network on ground truth haptics. Second, we provide the query in image space, we then employ $f_1$ to compute predicted haptics. We then used the predicted haptics to identify the object as seen in 4.1. We find that the network was able to predict the object identity using haptic forces on the known objects samples per category with near 100% accuracy. When employing the predicted means, this accuracy was a bit lower.

| Inputs | 11 Object Accuracy |
|---|---|
| Ground Truth Haptics | .99 |
| Predicted Haptics | .54 |

Table 1: Table showing object identification test accuracies for training objects given ground truth and predicted haptic forces. There are 11 objects in the training set, many of which were imported from ShapeNet (Chang et al. (2015)).

## 4.2 EFFECTIVE EXPLORATION LENGTH FOR CLASSIFICATION

In our current setup the agent employs a predetermined sampling routine to explore the object using tactile sensors. A natural question is how much exploration is required to accurately identify the object just using tactile information. To answer this, we trained separate tactile classification networks (i.e. $f_2$) using different number of samples obtained by exploration. Results presented in table 4.2 show that about 100 samples are sufficient for accurate classification and performance improves only marginally when 500 samples are collected. The inputs to the network still consisted of averaged haptic forces but they were computed over different episode lengths. Every object from Figure 3 were presented in various poses. During test time, held-out samples were presented.

| Episode Length | Accuracy |
|---|---|
| 1 | 9% |
| 10 | 14% |
| 100 | 93% |
| 500 | 95% |

Table 2: Table showing object identification accuracies given different episode lengths on the 11 objects imported from Shape-Net.

From Table 4.2 we see that when only one time step is used the classification accuracy is just over chance ($0.04\%$). This number increases significantly even with a few time steps and saturates fairly quickly after that.

## 4.3 IDENTIFYING NOVEL OBJECTS IN THE DARK

While the above experiments demonstrate that we are able to classify objects based on tactile sensing alone, they donot show if it possible to retrieve the object from interest when only an image of the object is presented (please see our experimental setup Figure 1).

An input image $I$ of the object to be retrieved is presented to the agent. A set of $p$ objects where $p \in K$ are presented to the hand. All the objects presented are novel. The model predicts the haptic response $\hat{\mathcal{H}}_t$ of the image using $\mathcal{F}_1$. We then use the predicted haptic response as an input to our classification network $\mathcal{F}_2$. Since $\mathcal{F}_2$ was not trained on the objects that we are trying to identify, we then identify the object using a nearest-neighbor classifier in the latent space of the network. We call this space *embedded haptics*. We compare the performance of this sampling classification by fitting the k-NN model with both raw haptic predictions and embedded haptic predictions for the $p$ objects presented.

We train $\mathcal{F}_1$ and $\mathcal{F}_2$ networks on 11 objects. During test time from the 14 novel objects, a set of $p$ objects are are out in the drawer (see Figure 1). The higher $p$ is, the harder the task is. The results of our method for $p = 2, 3, 5$ are presented in Table 4.3.

We generate a subset of held-out objects as well as three haptics templates per object. We then classify 1 query image from the set of objects with a kNN classifier with $k = 3$.

To compute the mean precision and standard deviation in this precision we run 20 classification trials. For each trial, we classify 5000 query images following the above procedure. Mean precision and standard deviations are then computed across these trials.

| # objects | Raw Hptx | Embedded Hptx |
|---|---|---|
| 2 | 58.5 % $\pm 0.7\%$ | 65.9 % $\pm 0.8\%$ |
| 3 | 42.5 % $\pm 0.8\%$ | 50.0 % $\pm 0.7\%$ |
| 5 | 26.6 % $\pm 0.5\%$ | 34.2% $\pm 0.6\%$ |

Table 3: Table showing object identification accuracies given 2, 3 and 5 held-out objects to be discriminated from. We see that the haptic embedding yields a meaningful increase in classification accuracy.

## 5 DISCUSSION

We present a model that when presented with a query image can identify a novel object from a set of $p$ objects using tactile sensing only. As the number of novel objects presented in a single trial increased, this task quickly became more challenging.

We show that a model with pre-determined exploration routine can identify the object but a richer exploration of the objects could allow us to answer more challenging inferences such as pose, texture, etc. One could imagine doing this by training a RL agent that uses the performance of $\{_2$ as a reward while generating trajectories that explore an object for identification.

Presently, we train our two networks $\mathcal{F}_1$ and $\mathcal{F}_2$ independently. Since, our larger goal is to identify objects it would be interesting to jointly optimize the network to maximize object identification. This can help smooth prediction errors that are not consequential to classification or identification.

While the MuJoCo simulator provides fast physics solvers that can compute realistic contact forces they only compute normal contact forces. Research in neuroscience has shown (Lederman & Klatzky (1993)) that a variety of forces are applied and measured while employing tactile sensing. It would be interesting to perform similar experiments on robot grippers equipped with tactile sensors.

Our current setup still trains network $\mathcal{F}_2$ in a supervised fashion. This is biologically implausible and tedious for practical scalability on real robots. It would be interesting if we posed this problem as a self-supervised problem and explored learning to identify novel objects using tactile sensing.

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
