# OpenReview forum: "Classification in the dark using tactile exploration"
_ICLR.cc/2019/Conference_

### Official Review · AnonReviewer1 · 2018-11-02
**Poorly written paper, with lots of confusing statements and incomplete description of the concepts and references introduced. Very weak contribution.**

**Rating:** 2
**Confidence:** 5

**Review:**

This paper is poorly written, and looks like it was not proof-read.
Presentation of the problem at hand is presented over so many times that it becomes confusing.
Authors ought to better define the image description space of the objects and the haptic space.
More interesting would have been a good explanation of the different sensors used in the anthropomorphic hand  and the vector built to represent the different sensed objects.
The most expected contribution of this work is barely explained: how the haptic sensors' values/object labels vectors were built and fed to the predictor network, what their values looked like for the various objects, how these vectors clustered for the various objects etc.

Among the many evident weaknesses:
- Domain specific concepts and procedures of most importance to this work are not explained: "... measure various physical properties of objects using the bio-tac sensor using five different exploration procedures (EP)".  Page 3, Paragraph 1. Bio-tac sensor and most importantly exploration procedures (EP) should be presented more clearly.
- Incomplete and out of nowhere sentences are common: "The SHAP procedure
was established for evaluating prosthetic hands and arms. With this idea in mind, prior work (?)
built a prosthetic arm which could ..." Page 4, Paragraph 1.
- Many references are not well introduced and justified: "We then trained the network using
ADAM (Kingma & Ba (2014)) with an initial learning rate set to 1e-4." Page 4, Paragraph 6. In the same paragraph,  authors explain using "The groundtruth predictions were per-channel averaged haptic forces" without having defined those channels (that one can guess but shouldn't). Concepts have to be clearly defined prior to their use.

---

### Official Review · AnonReviewer3 · 2018-11-04
**Not clear what's the contribution.**

**Rating:** 3
**Confidence:** 5

**Review:**

The authors propose a task of classifying objects from tactile signals. To do this, the image and haptic data are collected for each object. Then, image-to-haptic and haptic-to-label predictors are trained by supervised learning. In the experiment, prediction accuracy on unseen object class is studied.

The paper is clearly written although it contains several typos. The proposed task of cross-modal inference is an interesting task. I however hardly find any significance of the proposed method. The proposed method is simple non-end-to-end predictors trained by supervised learning. So, the proposed model seems more like a naive baseline. It is not clear what scientific challenge the paper is solving and what is the contribution. Also, the performance seems not impressive. I’m not sure why the authors average the haptic features. Lots of information will be lost during the averaging, why not RNNs. Overall, the paper seems to require a significant improvement.

---

### Official Review · AnonReviewer2 · 2018-11-04
**exciting research problem, but there are problems in the level of detail in the paper, and the approach taken is not convincing**

**Rating:** 4
**Confidence:** 3

**Review:**

This is an exciting research problem, and could be of broad interest in robotics.  The problem posed, and associated data sets and simulation code, if shared, could be an interesting and novel source of challenge to machine learning researchers.

However, the paper appears to be a fairly early work in progress, with missing detail in many areas, and making some odd approximations. One concern is the use of averaged haptic readings over a series of explorations, rather than the haptic reading for a specific pose. The approach of averaging seems certain to blur and confound things unnecessarily, making it harder for the system to learn the relationship between pose, object form and sensation.

The paper itself has weaknesses, e.g. on p5 you say "When employing the predicted means, this accuracy was a bit lower." when you actually have a drop from 99% to 54%! You do not specify which objects are used for this experiment. and in Section 4.2, you do not specify the exploration strategy used.

Can you clarify the explanation of the images in Figure 3 - you say that the image is as in Figure 3, but are you really giving the image of the object AND hand, or just the object itself (if so, you need to change the explanation).

---

### Meta-Review · Area_Chair1 · 2018-12-12
**unfinished**

**Confidence:** 5
**Recommendation:** Reject

**Metareview:**

The paper describes the use of tactile sensors for exploration.  An important topic which has been addressed in various previous publications, but is unsolved to date.

The research and the paper are unfortunately in a raw state.  Rejected unanimously by the reviewers, without rebuttal chances used by the authors.